# A Simplified 2D Numerical Simulation of Photopolymerization Kinetics and Oxygen Diffusion–Reaction for the Continuous Liquid Interface Production (CLIP) System

**DOI:** 10.3390/polym12040875

**Published:** 2020-04-10

**Authors:** Kentaro Taki

**Affiliations:** School of Mechanical Engineering, College of Science and Engineering, Kanazawa University, Kakuma machi, Kanazawa, Ishikawa 920-1192, Japan; taki@se.kanazawa-u.ac.jp; Tel.: +86-76-264-6257

**Keywords:** photopolymerization, conversion, oxygen inhibition, 3D printer

## Abstract

Additive manufacturing is a versatile technology for producing customized 3D products. In 2015, the Continuous Liquid Interface Production (CLIP) system was developed as a part of projection-type, UV-curable resin 3D printers. The CLIP system utilized the dead zone where oxygen inhibition occurs and prevents the UV-cured product from adhering to the UV illumination window. The CLIP system successfully produced complex shapes in a short time. This study investigated how the relationship between the photopolymerization rate, oxygen inhibition rate, and oxygen diffusion rate affects the shape of the product by means of a numerical simulation of the photopolymerization kinetics with oxygen diffusion and reaction. The results indicate that the vertical production speed and transmittance of UV light are crucial to controlling the conversion and shape precision of products.

## 1. Introduction

UV-curable resin is one of the most popular materials in additive manufacturing, also known as three-dimensional (3D) printing [1]. Indeed, the first type of additive manufacturing to be invented was stereolithography, in which a UV-curable resin in a vat is selectively solidified by a UV laser [2]. Another type of 3D printer is the inkjet-type printer; these are distinguished by their material saving properties compared with other printers. Inkjet-type printers can selectively place a UV-curable resin at a desired position, store the resin in the reservoir tank of the printer head, and use it for the next printing operation [3]. The third type of additive manufacturing that uses UV-curable resin is a projection-type printer. Illumination with UV or near-UV/visible light through the Digital Light Processing (DLP) Mirror^®^ (DLP-type projector) can selectively solidify the resin in a vat. An object can be printed by repeating the illumination and lifting of the object [3]. Recently, the projection-type printer has been improved to accelerate the printing time by positively employing the oxygen inhibition reaction, which is called Continuous Liquid Interface Production (CLIP) [4]. CLIP forms a dead zone where the “printing reaction”, or polymerization, does not occur because oxygen quenches the radicals. In a typical lifting-type 3D printer, the polymerized monomer adheres to the illumination window. To tear the product away from the window, the printer must lift the product up after every UV illumination, which is time consuming. The CLIP system can continuously lift the product up, whereas previous projection-type 3D printers sequentially repeat the illumination and lifting process on the object. Thus, the CLIP system successfully increases the production speed by 25 to 100 times as it can continuously lift the product. CLIP has been commercialized by Carbon 3D and is now applied in various applications.

Although the printing speed of other 3D printers has also improved, considerable time is still required to print objects. In addition, in 3D printers, often the shape of the printed object does not meet the as-designed shape because the resin shrinks. Measuring the size of the printed object, adjusting its shape, and printing it again requires a huge amount of time to obtain an object in the desired shape. If we can simulate the shape of the object on a computer, before actually printing the object, to determine its final shape under influences such as shrinkage, we can reduce the number of times the object is printed; thus, the time from the design step to object completion can be greatly shortened.

Very few studies have simulated the shaping process of 3D printers that use UV-curable resin. However, the photopolymerization kinetics in stereolithography [5] and in projection-type 3D UV printers [6] have been studied in detail using numerical simulations. In these types of 3D printers, the UV-curable resin is selectively photopolymerized in a vat. The layer-by-layer lamination and UV irradiation process in the 3D UV inkjet printer were simulated using an experiment and numerical calculations [7]. In that experiment, spin-casting and UV irradiation were sequentially combined, and a laminated layer consisting of two layers was produced. The distribution of the conversion of C=C bonds was measured using a confocal laser Raman microscope. To clarify the distribution of oxygen and to improve the geometrical resolution of conversion, the numerical calculation developed in the previous study [7] was also applied here. The numerical calculation model was modified to employ the rate coefficients of reactions among the initiator radical, C=C bonds, and oxygen, as well as the UV light attenuation and the conversion dependence of the diffusion coefficient. Recently, numerical simulations of the CLIP system have advanced dramatically. A coarse-grained molecular dynamics simulation was performed to improve the quality and accuracy of printed parts [8]. Further, machine learning using experimental datasets and physics-based simulations is applicable to deduce the optimal printing speed [9]. However, the printing of patterns, e.g., the checkerboard structure, has not yet been studied. This kind of structure is suitable for understanding the effects of oxygen diffusion and the transmittance of UV light on the product shape. 

In this study, a simplified numerical simulation model of CLIP was developed, wherein the effect of fluid flow induced by lifting the product was neglected. The model consisted of photopolymerization kinetics and the diffusion and reaction of oxygen in the UV-curable resin. The model parameters for the UV-curable resin of diurethane dimethacrylate/Irgacure 184 that were determined in our previous study [7] were used here to simulate the conversion and oxygen concentration distribution over time. In the numerical simulation, we focused on the conversion and concentration of the C=C bonds in the UV-curable resin because they are the essential factors in shaping the UV-cured resin. Initially, the number of converted C=C bonds is zero, and the resin is liquid. Then, conversion increases and the liquid becomes solid. The simplest way to understand how the printed shape is formed is by simulating and monitoring this conversion process. Oxygen plays a major role in the performance of the CLIP system. Therefore, the concentration of oxygen was monitored to understand its effects on the depth of the dead zone, where the oxygen inhibition reaction is more prominent than the polymerization of monomers, and on the extension of the UV-cured area. 

## 2. Theory and Implementation

The CLIP system lifts the photopolymerized product up at a constant velocity. While lifting the product, raw material around the product flows under the bottom of the product. Continuous lifting and flow of the raw material were enabled by the concept of the dead zone of the CLIP system. However, to simulate this system numerically, the chemical reaction, solidification, and fluid flow are difficult to couple because of the large differences in the characteristic time; ratio of the molar concentration of C=C bonds; the propagation rate coefficient, i.e., [*M*](mol/m^3^)/*k*_p_(m^3^/mol∙s); the ratio of product size and diffusion coefficient, *L*^2^(m^2^)/*D*(m^2^/s); and the ratio of viscosity and atmospheric pressure, μ(Pa∙s)/*P*_a_(Pa); which have values of ~10, 10^2^, and 10^-4^, respectively. For simplicity, this study focuses on the chemical reaction, i.e., photopolymerization, oxygen inhibition, and the diffusion of oxygen.

### 2.1. Photopolymerization Kinetics

The numerical simulation of photopolymerization kinetics was studied by means of integration of ordinal or partial differential equations of the initiation, propagation, termination, and oxygen inhibition reactions [7,10]. The following equations were recently reported for the photopolymerization of diurethane dimethacrylate with 1-hyroxy-cyclohexy phenyl ketone [7].

Photodissociation rate of the photoinitiator:(1)$$R_{i}=\varphi \varepsilon [\mathrm{PI}]I(z),$$

Initiator’s radical:(2)$$\frac{d[\mathrm{IR}]}{dt}=R_{i}-k_{i}[\mathrm{IR}][M]-k_{\mathrm{io}}[\mathrm{IR}][O_{2}],$$

C=C bond:(3)$$\frac{d[M]}{dt}=-k_{i}[\mathrm{IR}][M]-k_{p}[\mathrm{MR}][M],$$

Macro radical:(4)$$\frac{d[\mathrm{MR}]}{dt}=k_{p}[\mathrm{PR}][M]-k_{t}{\left[\mathrm{MR}\right]}^{2}-k_{O}[\mathrm{MR}][O_{2}],$$

Oxygen inhibition:(5)$$\frac{\partial [O_{2}]}{\partial t}=\frac{\partial }{\partial z}\left(D_{O}\frac{\partial [O_{2}]}{\partial z}\right)+\frac{\partial }{\partial x}\left(D_{O}\frac{\partial [O_{2}]}{\partial x}\right)-\left(k_{\mathrm{io}}[\mathrm{IR}]+k_{O}[\mathrm{MR}]\right)[O_{2}],$$

Conversion dependence of diffusion coefficient of oxygen:(6)$$D_{O}=D_{O0}\exp\left(-\alpha /f\right),$$
where *R*_i_ is the photodissociation rate of the photoinitiator, φ (0.8) is the quantum yield, ε (1.717 m^3^/mol/m) is the molar absorption coefficient, [PI] (54.55 mol/m^3^) is the molar concentration of photo initiator, and *I*(*z*) is the UV intensity at the vertical position of *z*. In addition, [IR] is the initiator’s radical molar concentration, *k*_i_ (2.5 × 10^4^ m^3^/mol∙s) is the initiation rate coefficient, [M] (initial concentration 4.64 × 10^3^ mol/m) is the molar concentration of C=C bonds, *k*_io_ (5.4 × 10^6^ /mol∙s) is the oxygen inhibition reaction rate coefficient between the initiator’s radical and oxygen, [O_2_] (initial concentration 1.3 × 10^−2^ mol/m^3^) is the oxygen concentration, *k*_p_ (initial value 1.145 × 10^5^ L/mol∙s) is the propagation reaction rate constant, [MR] is the macro radical concentration, *k*_O_ (5 × 10^5^ m^3^/mol∙s) is the oxygen inhibition reaction rate constant between the macro radical and oxygen, *D*_O_ is the mutual diffusion coefficient of oxygen, *D*_O0_ (1.08 × 10^−10^ m^2^/s) is the diffusion coefficient of oxygen in monomer, α (0.358) is a parameter of reduction of the diffusion coefficient, and *f* is the free volume fraction. As in previous studies [7,10], the depression of the rate coefficient of propagation and termination was considered using the Anseth–Bowman model [11]. 

### 2.2. Implementation

Figure 1 shows the geometry of the numerical calculation. The depth and width were 200 and 400 μm, respectively, and were discretized by 5 μm. The initial gap was 50 μm, and the UV light source was moved downward at constant velocity. 

The vertical position of the UV light source is expressed as
(7)$$b=H_{0}+ut,$$
where *H*_0_ (50 μm) is the initial gap, *u* (0.1 mm/s) is the “lift-up” speed or downward velocity of the UV light source, and *t* is time. The UV light was attenuated according to the Lambert–Beer law in the *z* direction:(8)$$E\left(x,z\right)=E_{0}\left(x\right)10^{-\varepsilon [PI]\left(b-z\right)}\;\left(z\leq b\right),$$
where *E*_0_(*x*) is the intensity of UV light depending on position *x.* As shown in Figure 1, *E*_0_(*x*) was set to 10 mW/cm^2^ in a range from 40 to 80 μm and from 120 to 160 μm. Other than in this area, *E*_0_(*x*) was zero. ϵ is the molar absorption coefficient, [PI] is the molar concentration of the photoinitiator, *b* is the vertical position of UV light. The UV intensity below the UV light source was zero:(9)E(x,z)=0 (z>b),

The periodic boundary condition was applied to the concentration of oxygen in the horizontal direction:(10)$$[O_{2}]\left(0,z,t\right)=[O_{2}]\left(W,z,t\right),$$

Simultaneously, the oxygen permeation line moved downward at the same speed as the UV light source. It was assumed that the permeation rate of oxygen through the window was sufficiently fast. The concentration of oxygen at the UV light source was constant: (11)$$[O_{2}]\left(x,b,t\right)={\left[O_{2}\right]}_{0},$$

The oxygen did not diffuse away from the top:(12)$${\frac{\partial [O_{2}]}{\partial z}|}_{z=0}=0$$

The partial differential equation of the diffusion–reaction equation of oxygen (Equation (5)) was discretized explicitly and integrated as a set of ordinal differential equations. The numerical simulation was implemented using a home-made MATLAB code with the *ode15* integrator. The parameters used to perform the simulation were determined with a mixture of diurethane dimethacrylate and 1-hydroxy cyclohexyl phenyl ketone, similar to the approach used in our previous studies [7,10].

## 3. Results and Discussion

The numerical simulation of C=C bond conversion as well as oxygen concentration was performed at the lift-up speed of 0.1 mm/s. A video of the change of the C=C bond and oxygen concentration is shown in the Appendix A. Figure 2 shows a contour plot of the normalized oxygen concentration and C=C bond conversion distribution. Figure 3 shows the normalized oxygen concentration and C=C bond conversion in the vertical direction at *x* = 60 μm. Figure 4 shows the change in the contour plot of the oxygen concentration, 0.9, and C=C bond conversion, 0.1. On the basis of these figures, the change in oxygen concentration and C=C bond conversion of the CLIP system is explained. 

The horizontal white line is the position of the UV light. The UV light moved downward and was emitted upward. The initial normalized concentration of oxygen was 1.0 in UV-curable resin. Then, the UV light was irradiated to the UV-curable resin. The UV light dissociated the photoinitiator and produced an initiator radical. The initiator radical reacted with the dissolved oxygen. The oxygen concentration decreased with time. As shown at t = 0.10 s, the contour plot of the oxygen concentration equaled 0.9, appearing as a dotted line. After 0.25 s, the interior of the dotted line expanded and its color became light. This indicates that the C=C bond conversion started to increase, and the propagation reaction started as the C=C bond group was consumed by the radicals. From 0.50 to 1.25 s, the interior of the dotted line expanded and the C=C bond concentration increased with time, and the UV light moved downward. 

A dead zone existed between the position of the UV light, indicated by horizontal arrows in Figure 4, and the C=C bond conversion was larger than 0.1. The vertical position of the UV light coincides with the lowest part of the oxygen concentration contour line. The oxygen concentration distribution along the vertical direction shown in Figure 3 indicates that the concentration gradient above 0.10 s is steep. The oxygen concentration decreased sharply near the UV light. The distance between the position of the UV light and the lowest position of C=C bond conversion, 0.9, is plotted in Figure 5. The distance was constant, while these positions increased linearly with time. Clearly, this distance must be the dead zone. The size of the dead zone did not change with time as the UV light intensity and oxygen permeation rate were constant. 

The lift-up speed is the major concern related to the production speed. Figure 6 shows the snapshots of a lift-up speed of 1 mm/s. Although this speed is extraordinarily high, it was used to clarify the effect of lift-up speed on the dead zone and C=C bond conversion. As shown in Figure 6, the snapshots were taken 10 times faster than those in Figure 2. From t = 0 s to 0.03 s, the contour plot of the oxygen concentration (0.9) expanded downward. At 0.07 s, it appeared that the C=C bond conversion started. Then, the C=C bond conversion increased and expanded with an increase in time. Obviously, the dead zone, where the vertical distance from the position of the UV light and C=C bond conversion becomes larger than zero, is expanded by an increase in lift-up speed. The magnitude of the C=C bond conversion became much lower than that of 0.1 mm/s. The maximum C=C bond conversion at 1 mm/s was 0.016, while that at 0.1 mm/s was 0.8. This is because the exposure time was shortened and the UV light source was moved away. The faster lift-up speed resulted in a reduction in the C=C bond conversion. 

Next, a simulation was carried out to make a lattice-shaped pattern by changing the position to be irradiated with UV light from moment to moment. First, *E*_0_ (*x*)= 10 mW/cm^2^ for the area of 40 ≤ *x* ≤ 80 and 120 ≤ *x* ≤ 150 μm. For the other area, i.e., 0 ≤ *x* < 40, 80 < *x* < 120, and 150 < *x* < 200 μm, *E*_0_(*x*) = 0 mW/cm^2^. The area irradiated with UV light and the area not irradiated were alternately switched every 100 ms.

As the area irradiated by UV light was polymerized, it was expected that a lattice pattern of polymerized and nonpolymerized areas would be formed by the UV light which was switching and moving vertically. However, as shown in Figure 7, contrary to expectations, the distribution of the C=C bond conversion did not change to a lattice shape, even when switching the irradiation position of the UV light every 100 ms. This was caused by the penetration depth of UV light through the UV-curable resin. The photoinitiator used had a relatively low molar absorption coefficient. The area of UV irradiation was switched every 100 ms. 

From *t* = 0 to 0.1 s, the UV light irradiated to Areas a and c. The oxygen concentration decreased and the contour lines of oxygen concentration (0.9) appeared. In the next 0.1 s, Area b was exposed to the UV light. Then, the oxygen was consumed and the contour line appeared. For Areas a and c, the contour line moved upward from the UV light position due to the diffusion of oxygen. In the next 0.1 s, Areas a and c were exposed to the UV light again. The contour line of oxygen met the UV light position and Area b moved backward. The switching of the UV light position continued every 0.1 s for 1 s. As shown at *t* = 0.3 s, the C=C bond conversion of Areas a and c advanced rather than Area b, because these areas were exposed first. The conversion of the C=C bond further advanced when the area was exposed to the UV light. Even though the exposition was switched, the conversion distribution did not show a lattice shape as the UV light penetrated deeply. To make a lattice shape, the penetration depth of UV light needs to be shortened.

To shorten the penetration depth, a model of the UV absorber was added to the original equation of UV light attenuation of Equation (8). The UV light was observed by the UV absorber and limited to penetrate deeply. The molar concentration of the UV light absorber was 100 times larger than that of the photoinitiator. A series of snapshots is shown in Figure 8. From *t* = 0.00 to 0.10 s, Areas a and c were exposed to UV light. Two closed areas of contour lines appeared as the penetration depth of the UV light shortened. In the center of each two areas, the C=C bond conversion advanced as the color red became bright. From *t* = 0.10 to 0.20 s, Area b was exposed. Interestingly, the closed areas of the contour lines were connected by the bridging of Area b. Then, the C=C bond conversion started to advance.

Figure 9 shows the series of snapshots up to *t* = 1.20 s for every 0.2 s. A lattice shape of the C=C bond conversion appears clearly as a result of switching the exposed area to the UV light. However, the magnitude of the conversion is far lower than the results in Figure 7. As the UV absorber was used, the UV light intensity to the photoinitiator reduced by 1/100 times. The initiation reaction rate became much slower. To realize the high conversion as well as a short penetration depth, it is essential to choose a photoinitiator that has a higher molar absorption coefficient and a higher production efficiency for the initiator radical.

## 4. Conclusions

In this study, a numerical simulation of a CLIP system was performed. The numerical simulation model was based on a previous study regarding layer-by-layer inkjet 3D printing using UV-curable resin. For the sake of simplicity, the fluid motion was neglected, and the vertical movement of the UV light was considered instead of a realistic motion of the lifting of the photopolymerized parts. Despite this simplification, the numerical simulation results showed the characteristic properties of a CLIP system and where the dead zone and continuous printing were realized. The lift-up speed is a crucial factor to determine the length of the dead zone and C=C bond conversion. A UV absorber is required to shorten the light passing through the resin and improve the reproduction of the designed shape.

In this study, the temperature of the UV-curable resin was assumed to be constant for simplicity. However, the actual photopolymerization emits heat, and the temperature of the UV-curable resin increases autocatalytically. The uneven temperature distribution makes the UV-curable resin flow in an unintended direction. Heating and cooling the UV-cured resin by photopolymerization stores the thermal stress in the resin. The numerical simulation of photopolymerization kinetics is not straightforward because the temperature dependence of the kinetic constants is difficult to determine. Real-time Fourier Transform Infrared (FT-IR) Spectroscopy is a nonisothermal measurement. Although photo differential scanning calorimeter (DSC) is an isothermal measurement, the rapid kinetics of photopolymerization limit the ability of this method to obtain accurate data. In future studies, I will resolve these issues and realize the nonisothermal simulation of 3D printing in the CLIP system.

## Figures and Tables

**Figure 1 polymers-12-00875-f001:**
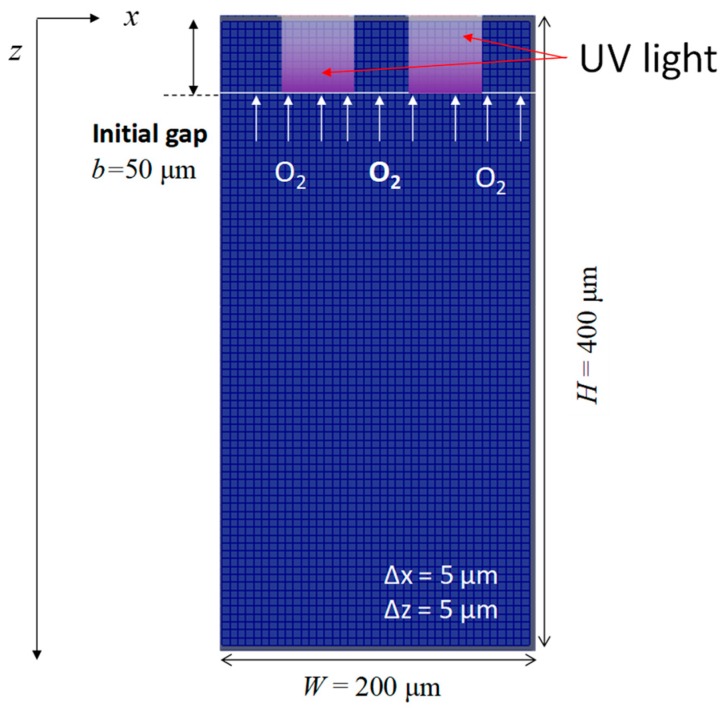
Geometry of numerical simulation.

**Figure 2 polymers-12-00875-f002:**
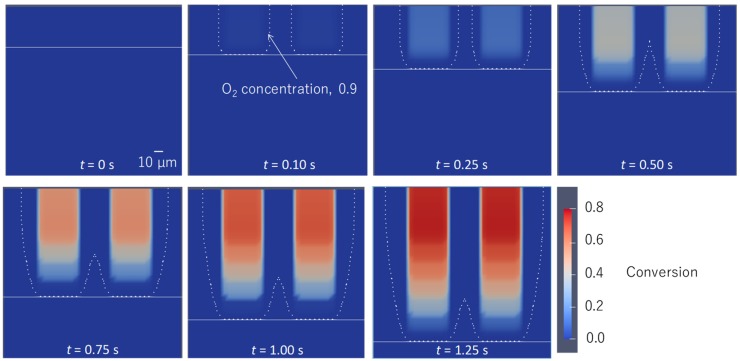
Contour plot of normalized oxygen concentration of 0.9 and C=C bond conversion distribution at a lift-up speed of 0.1 mm/s. The corresponding video is Video S1 on the website of the Appendix A.

**Figure 3 polymers-12-00875-f003:**
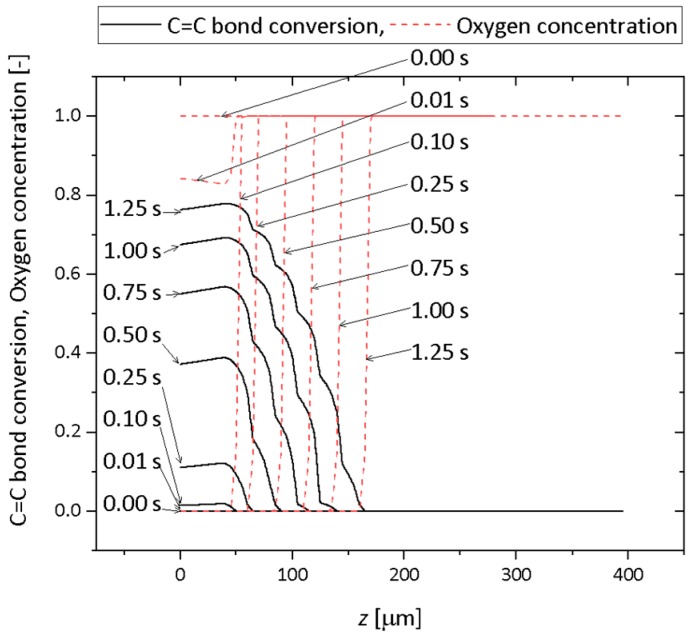
Contour plot of normalized oxygen concentration of 0.9 at a lift-up speed of 0.1 mm/s. The corresponding video is Video S1 on the website of the Appendix A.

**Figure 4 polymers-12-00875-f004:**
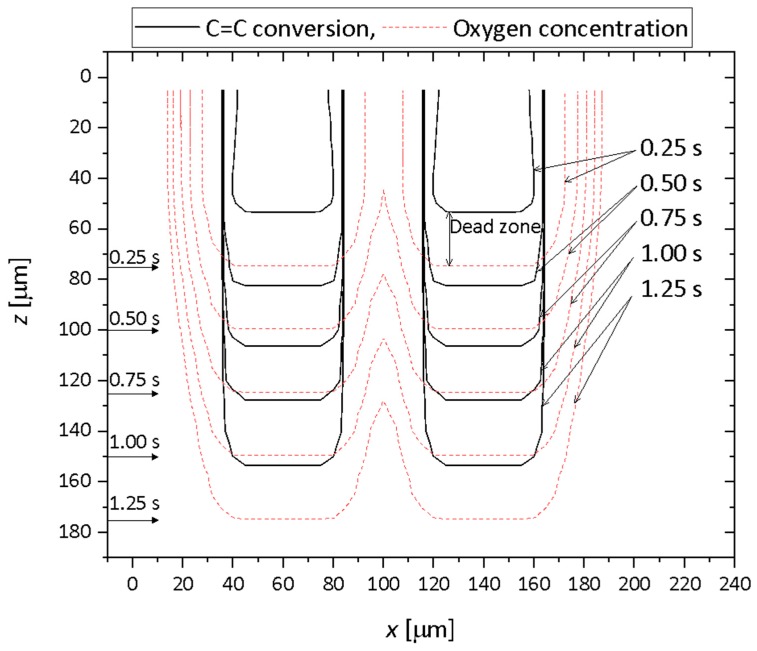
Superposition of the contour plot of normalized oxygen concentration of 0.9 and C=C bond conversion as well as UV light position at a lift-up speed of 0.1 mm/s. The horizontal arrows on the left side are the position of UV light for each time point.

**Figure 5 polymers-12-00875-f005:**
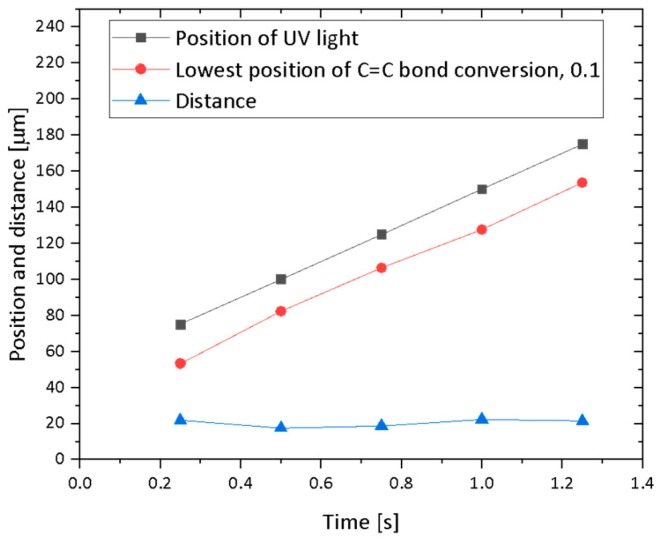
Position of UV light, the lowest position of C=C bond conversion (0.1), and their distance.

**Figure 6 polymers-12-00875-f006:**
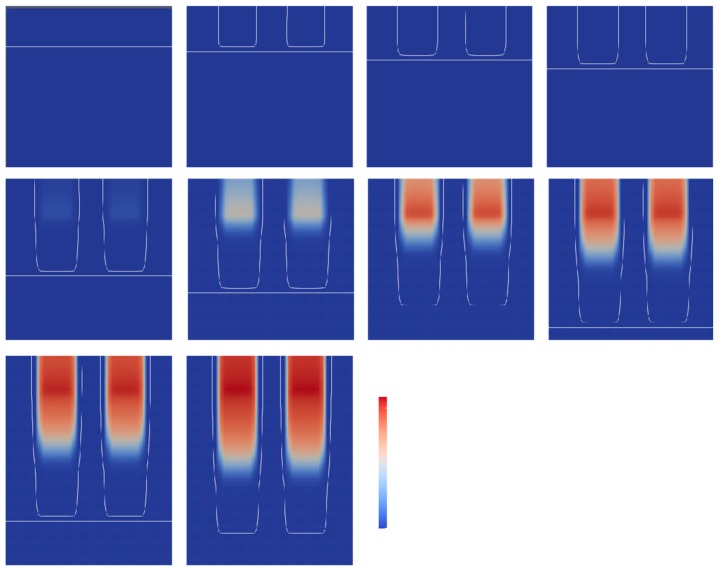
Contour plot of normalized oxygen concentration of 0.9 and C=C bond conversion distribution at a lift-up speed of 1 mm/s. Notice: the color bar of conversion has a different range from that in Figure 2. The maximum range of the color bar is 50 times smaller than that in Figure 2.

**Figure 7 polymers-12-00875-f007:**
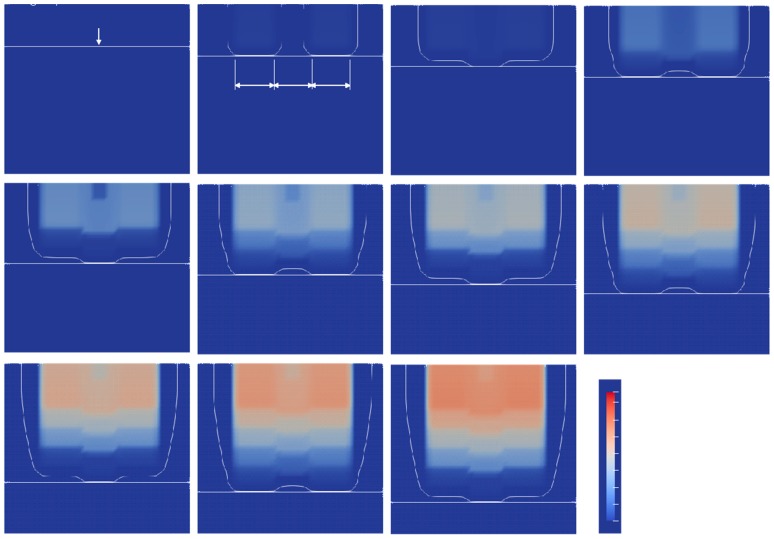
Contour plot of normalized oxygen concentration of 0.9 and C=C bond conversion distribution at a lift-up speed of 0.1 mm/s and alternating UV light position.

**Figure 8 polymers-12-00875-f008:**
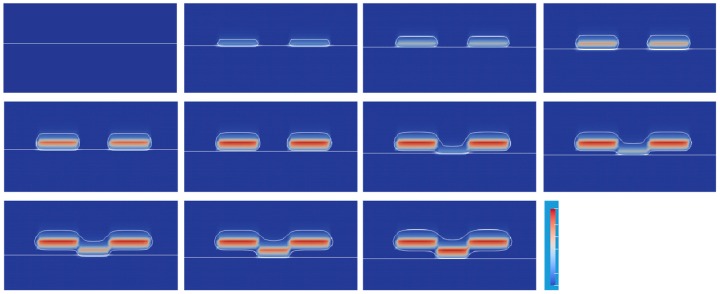
Contour plot of normalized oxygen concentration of 0.9 and C=C bond conversion distribution at a lift-up speed of 0.1 mm/s and alternating UV light position with UV absorber.

**Figure 9 polymers-12-00875-f009:**
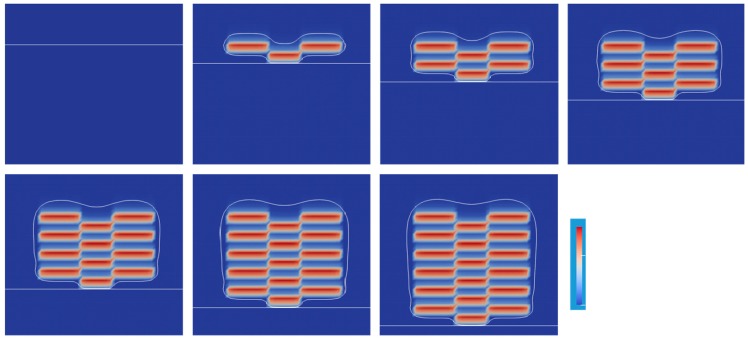
Change of oxygen and C=C bond conversion at the lift-up speed of 1 mm/s. The corresponding video is Video S2 on the website of the Appendix A.

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
