# Peer review of "A Simplified 2D Numerical Simulation of Photopolymerization Kinetics and Oxygen Diffusion–Reaction for the Continuous Liquid Interface Production (CLIP) System"

_polymers, 2020, doi:10.3390/polym12040875_

Round 1
Reviewer 1 Report
This work brings an interesting insight into the kinetics of the Continuous Liquid Interface Production 3D printing. The author has developed a numerical model consisted of photopolymerization kinetics and oxygen diffusion reaction in the liquid precursor.
Below there are few of my suggestions to the author.
1) line 81-82, the values need units and symbols explanations.
2) Sentence 87-88 needs a reference.
3) Is in eq12 should be D? dD/dz?
4) 146: The UV light moved to downward and the UV light was emitted to upward. This is confusing. Can this be re-formulated?
5) 157: mean Figure 4, not 3, right? Also, I think it would be very informative if you show the deadzone on one of the figures.
6) 211: should be area "b".
7) In title: CLIP stands for Continuous Liquid Interface Production.
Author Response
Dear reviewer and editor,
Thank you so much for your comments and suggestions. I agree with your suggestions, and I have appropriately corrected my manuscript accordingly. I hope it will be suitable for publication.
Best regards,
Kentaro Taki
1) line 81-82, the values need units and symbols explanations.
The units of symbols have been added as follows:.
[M] (mol/m3) /kp (m3/mol∙s), the ratio of product size and diffusion coefficient, L2 (m2) / D (m2/s), the ratio of viscosity and atmospheric pressure, μ (Pa∙s)/Pa (Pa).
2) Sentence 87-88 needs a reference.
Reference [7] was added.
3) Is in eq12 should be D? dD/dz?
It was corrected to be
∂[O2]/∂z|z=0=0
4) 146: The UV light moved to downward and the UV light was emitted to upward. This is confusing. Can this be re-formulated?
I agree with you. My visualization scheme is different from the real one. We have tried to visualize our system to be similar to the real CLIP system. However, it was not realized. We have thus concluded that the UV light moving downward and emitting UV light upward is the best way to represent our results.
5) 157: mean Figure 4, not 3, right? Also, I think it would be very informative if you show the deadzone on one of the figures.
It should be Figure 4 instead of Figure 3. The label “Dead zone” was added to Figure 4.
6) 211: should be area "b".
As shown at t = 0.3 s, the C=C bond conversion of area a and c advanced rather than area c …
“c” is replaced with “b”.
7) In title: CLIP stands for Continuous Liquid Interface Production.
I apologize; I have inserted the word “Interface“ in the title and where appropriate in the manuscript.
Reviewer 2 Report
The paper deserves publication. The proposed method is interesting and well described, the results presented seem comforting. My only concern is related to a (perhaps apparent) lack in deepening the state of the art. The author statement between lines 45-48, seems to me rather obvious, even considering that nowadays the design processes and software, which allow the simulation of the shape of the final product and the optimization of the production process, aimed at obtaining the desired shape, are already widespread on the market.
I suggest the author to deepen the literature review (perhaps increasing and updating the number of bibliographic references) and to compare his work with the methods already existing on the market and to highlight the fundamental differences between the proposed method and what is already in use.
This can help emphasize the merit of this work.
The text is legible, I have found a few typing errors that the author can fix quickly. In the abstract, between lines 15 and 18, the author has forgotten the verb that would allow us to understand the purpose of the paper, I suggest re-reading this period and correcting it appropriately.
On line 81 I suggest removing the space between / and D.
On line 244 the word "despite" uses a different font.
Author Response
Dear reviewer and editor,
Thank you so much for your comments and suggestions. I agree with your suggestions, and I have appropriately corrected my manuscript accordingly. In particular, in the Introduction section, I cited two new articles and pointed out issues with my manuscript.
I hope it will be suitable for publication
Best regards,
Kentaro Taki
I suggest the author to deepen the literature review (perhaps increasing and updating the number of bibliographic references) and to compare his work with the methods already existing on the market and to highlight the fundamental differences between the proposed method and what is already in use.
This can help emphasize the merit of this work.
I added two papers recently published and valuable to highlight and I made my issue clearer as …
Recently, numerical simulations of the CLIP system have advanced dramatically. A coarse-grained molecular dynamics simulation was performed to improve the quality and accuracy of printed parts [8]. Further, machine learning using experimental datasets and physics-based simulations is applicable to deduce the optimal printing speed [9]. However, the printing of pattern, e.g., the checkerboard structure, has not been studied yet. This kind of structure is suitable for understanding the effects of the oxygen diffusion and the transmittance of UV light on the product shape.
- Wang, Z.; Liang, H.; Dobrynin, A.V. Computer simulations of continuous 3-D printing. Macromolecules 2017, 50, 7794-7800, 10.1021/acs.macromol.7b01719.
- He, H.; Yang, Y.; Pan, Y. Machine learning for continuous liquid interface production: Printing speed modelling. J. Manuf. Syst. 2019, 50, 236-246, https://doi.org/10.1016/j.jmsy.2019.01.004.
The text is legible, I have found a few typing errors that the author can fix quickly. In the abstract, between lines 15 and 18, the author has forgotten the verb that would allow us to understand the purpose of the paper, I suggest re-reading this period and correcting it appropriately.
I added verb “affect” and removed “on”.
This study was performed to elucidate how the relation between the photopolymerization rate, oxygen inhibition rate, and oxygen diffusion rate affects on the shape of the product by means of a numerical simulation of photopolymerization kinetics with oxygen diffusion and reaction
On line 81 I suggest removing the space between / and D.
The space was deleted, as the reviewer suggested.
On line 244 the word "despite" uses a different font.
The font of “despite” was corrected. Thank you for pointing this out.